# Automatic Assessment of AK Stage Based on Dermatoscopic and HFUS Imaging—A Preliminary Study

**DOI:** 10.3390/jcm13247499

**Published:** 2024-12-10

**Authors:** Katarzyna Korecka, Anna Slian, Adriana Polańska, Aleksandra Dańczak-Pazdrowska, Ryszard Żaba, Joanna Czajkowska

**Affiliations:** 1Department of Dermatology, Poznan University of Medical Sciences, 61-701 Poznań, Poland; apolanska@ump.edu.pl (A.P.); aleksandra.danczak-pazdrowska@ump.edu.pl (A.D.-P.); rzaba@ump.edu.pl (R.Ż.); 2Department of Biomedical Informatics and Artificial Intelligence, Silesian University of Technology, 41-800 Zabrze, Poland; jczajkowska@polsl.pl

**Keywords:** artificial intelligence, actinic keratosis, high frequency ultrasonography, dermatoscopy, digital imaging

## Abstract

**Background:** Actinic keratoses (AK) usually occur on sun-exposed areas in elderly patients with Fitzpatrick I–II skin types. Dermatoscopy and ultrasonography are two non-invasive tools helpful in examining clinically suspicious lesions. This study presents the usefulness of image-processing algorithms in AK staging based on dermatoscopic and ultrasonographic images. **Methods:** In 54 patients treated at the Department of Dermatology of Poznan University of Medical Sciences, clinical, dermatoscopic, and ultrasound examinations were performed. The clinico-dermoscopic AK classification was based on three-point Zalaudek scale. The ultrasound images were recorded with DermaScan C, Cortex Technology device, 20 MHz. The dataset consisted of 162 image pairs. The developed algorithm includes automated segmentation of ultrasound data utilizing a CFPNet-M model followed by handcrafted feature extraction. The dermatoscopic image analysis includes both handcrafted and convolutional neural network features, which, combined with ultrasound descriptors, are used in support vector machine-based classification. The network models were trained on public datasets. The influence of each modality on the final classification was evaluated. **Results:** The most promising results were obtained for the dermatoscopic analysis with the use of neural network model (accuracy 81%) and its combination with ultrasound scans (accuracy 79%). **Conclusions:** The application of machine learning-based algorithms in dermatoscopic and ultrasound image analysis machine learning in the staging of AKs may be beneficial in clinical practice in terms of predicting the risk of progression. Further experiments are warranted, as incorporating more images is likely to improve classification accuracy of the system.

## 1. Introduction

Actinic keratoses (AKs) are pink, scaly plaques that usually occur on sun-exposed areas in elderly patients with fair skin phototypes. The rate of progression of a single AK lesion to squamous cell carcinoma (SCC) is estimated at around 0.1–20% [1,2]. In contrast, a higher number of lesions (>10) increases the patient’s cumulative risk of malignancy by up to 14% over the next 5 years [2]. Therefore, treating actinic damage is crucial in preventing transformation into an invasive lesion.

Dermatoscopy is the most frequently used fast auxiliary diagnostic tool, which remains the standard method to assess clinically suspicious lesions [3], and it is known to increase the sensitivity and specificity of diagnosing melanomas (MM) and non-melanoma skin cancer (NMSC) [4,5,6]. 

Ultrasonography is a non-invasive diagnostic method widely used in medicine, including dermatology, where visualization of the skin layers can be obtained using higher than usual frequencies (>18 MHz, high-frequency ultrasonography, HFUS). HFUS was primarily applied in dermato-oncology, where its relationship with tumor infiltrate thickness, especially in MM and basal cell carcinoma (BCC), correlated with histopathological examination [7]. Additionally, various other indications were presented recently, including monitoring inflammatory skin and neoplastic skin diseases [8,9,10,11,12,13,14,15,16]. There are no specific ultrasonographic features related to the type of skin tumor, and all may manifest as anechoic or hypoechoic lesions, with the possibility of subepidermal low echogenic band formation underneath the entry echo (known as SLEB). Due to the scale presence, SCC and AKs are the most difficult to assess with the use of HFUS, and limited data on HFUS in this indication are available in the literature. Single studies used the HFUS to evaluate treatment efficacy [17,18,19] and described its morphological features without relation to AK staging [20]. There is a lack of studies on HFUS that address the stage of the AK in relation to dermatoscopic findings, which may be helpful from a clinical point of view in treatment planning as well as in assessing progression or response.

### Machine Learning Application in Dermatology

Machine learning (ML) has many applications in various medical fields [21]. As a field primarily based on visual perception, dermatology can benefit from tools utilizing artificial intelligence (AI) [22]. It can be applied directly to assess the skin’s condition [23], detect tumors, segment lesions, and feature extraction [24,25,26].

Much attention has been paid to ML systems supporting the diagnosis of skin diseases, particularly cancerous lesions. Computer-assisted diagnosis accelerates the doctor’s work, whether by tools enabling faster image segmentation, data labeling, or direct data analysis and classification [25]. Convolutional neural networks (CNNs) simulate the operation of biological neurons and are currently highly efficient tools used for pattern recognition in image data.

Promising results are achieved, particularly in classifying malignant lesions such as MM. For this purpose, the lesion area is usually segmented from the surrounding skin [26]. Popular approaches consider handcrafted features, such as texture or morphological properties [27,28,29,30], and the use of artificial neural networks for feature extraction and classification [29,30,31].

Many ML algorithms target HFUS image analysis [32]. For skin layer segmentation, researchers often use U-shaped architectures consisting of an encoder and decoder module and their modifications [32,33,34]. Promising HFUS classification results in inflammatory skin diseases were reported for architectures like DeepLab v3+, ResNet, and Xception [34] and computational techniques like active contouring [35], level-set segmentation [36], or fuzzy C-means clustering [32].

HFUS images have been used to diagnose skin lesions of a malignant nature in [14,37,38,39]. In addition, HFUS can be used to assess underlying inflammation in such conditions as atopic dermatitis [8,12,40,41], psoriasis [42,43], and scleroderma [44] and others [45,46,47]. Useful features include the thickness and shape of individual layers [48], echogenicity [48,49,50], shape parameters of lesions [20], or parameters related to skin texture [51].

Most research on computerized AK assessment focuses on diagnosing AK against other conditions [52,53]. In [54], an original neural network (NN), was proposed to assist with clinical image segmentation of AK. Li et al. [55] focused on using AI in the auxiliary diagnosis of AK, that is, the distinction between AK and non-actinic keratosis (NAK). The authors compared popular architectures and showed that using modified MobileNet resulted in the best accuracy in binary classification (AK vs. NAK).

In our study, being the first in this area, we introduce an ML-based framework for automatic assessment of the ultrasonographic AK stage in relation to dermatoscopy. The developed methods enable fully automated segmentation of interesting skin layers in HFUS images using a fine-tuned convolutional network model. The segmentation results are used for handcrafted feature selection, which, combined with features extracted from dermatoscopic images, enables accurate AK assessment.

## 2. Materials and Methods

### 2.1. Clinical Study Design

This prospective study was performed on 54 patients presenting with AK lesions at the outpatient clinic at the Department of Dermatology, Poznan University of Medical Sciences. The patients were 53 to 89 (median age 74, 74% males, Fitzpatrick I–II). Clinical, dermoscopic, and HFUS examinations were performed on every subject. The patients did not receive any treatments prior to their visit. The clinico-dermatoscopic classification of each AK was based on a three-point Zalaudek scale [56] with the use of a smartphone (iPhone 8, Apple, Cupertino, CA, USA) camera coupled to a dermatoscope (DermLite DL5, Aliso Viejo, CA, USA, 10× magnification, dermatoscopy images sized 3024 × 4032 pixels). The data were analyzed by two experienced dermatologists based on dermatoscopic examination, HFUS results, and medical interview. The labels given by the council of experts referred to the degree of AK based on the clinico-dermatoscopic classification and assigned as stage/grade one, two, or three. 

In grade one, a pink or red pseudonetwork pattern with discrete white scales was usually noticeable. Grade two was characterized by an erythematous background alongside white or yellow, keratinized, and wide hair follicles (commonly known as the “strawberry pattern”). In grade three, wide hair follicles filled with keratinized masses on a scaly, white-yellow background were described [56]. In this stage, hyperkeratosis was also observed, usually as whitish, yellow, and structureless areas.

All the HFUS images of AKs were recorded with DermaScan C, a Cortex Technology, Aalborg, Denmark device, and a linear 20 MHz probe. The 1024 × 224 pixels images were cropped to 512 pixels for further analysis. The dataset collected for the present study consisted of 162 pairs of dermatoscopic photos and HFUS scans of AK: 92 stage one, 42 stage two, and 27 stage three. Single patient records include various AK stages, and the image data are collected from different skin areas. Three exemplary image pairs, i.e., dermatoscopic and HFUS images of the same lesion recorded during the examination, are shown in Figure 1. 

### 2.2. Methods

The AI framework for multimodal image processing is visualized in Figure 2. It combines deep models: CFPNet-M for HFUS image segmentation and EfficientNet for feature extraction with SVM for final AK assessment. Individual steps are interspersed with classic data analysis tools (handcrafted feature extraction) in places where due to insufficient training data it was impossible to use deep models. The algorithm works double tracks. The first targets dermatoscopic image analysis, whereas the second focuses on HFUS images. All the extracted features are combined and used for final data classification. The individual steps of the analysis are described in the following sections. The algorithms taken into account at each stage of the analysis are listed in Appendix A, Table A1.

#### 2.2.1. Feature Extraction

***Skin layers segmentation in HFUS images*:** The first step in processing the ultrasound images was to extract the skin layers of interest (entry echo layer and SLEB). Based on current research in this field, four U-shaped architectures [57,58,59,60] were considered for the segmentation task. The finally chosen CFPNet-M, proposed by Lou et al. [57], utilizes a contextual feature pyramid module consisting of multiple convolutional layers with different coefficients to capture features on different perceptual fields. The outputs of those convolution layers are then combined and processed further downstream in the model, allowing the network to pick both local and global context information [57]. The authors highlight the model’s application in biomedical image segmentation, as it achieves similar results to more elaborate and memory-intensive models, making it possible to apply it to real-time systems [57].

Due to the limited number of images collected, a transfer learning method was used. Two HFUS datasets of inflammatory skin diseases were selected for pre-training: the first one, consisting of 380 images, was recorded with the DUB SkinSkanner75 device, Denmark device, 75 MHz probe [61], while the second, consisting of 158 images, was recorded with the DermaScan C, Cortex Technology device, equipped with a 20 MHz linear probe. The model was then fine-tuned using 10-fold cross-validation with the dataset collected for this study. The technical details of model training parameters are in Appendix B (Table A2). 

***Dermatoscopic image processing*:** Round corner frames and measurement marks are often visible on dermatoscopic images. Due to the specific location of AK lesions, e.g., in the area of the patient’s ear or nose, a blurred background may also be present (see Figure A1 in Appendix B). These elements do not carry useful information, so they were removed before feature extraction. An algorithm combining image intensity adjustment, thresholding, and fuzzy c-means clustering was utilized to segment those parts.

The following step is removing hairs that do not carry valuable information about the skin condition and can influence further classification. Both dark and light hairs, as well as the ruler, were extracted using the modified DullRazor algorithm [62] (the parameters are provided in Appendix B), and the in-painting technique was applied to smooth the images. 

***Handcrafted features*:** HFUS image characteristics were calculated based on the binary masks obtained as a result of automatic segmentation. The identified morphological features were perimeter-to-area ratio and depth descriptors. An important advantage of morphological features is that they are independent of the type and settings of the ultrasound camera, which will affect textural features [63]. 

On the other hand, texture features allow the evaluation of the properties of the objects, referring to the spatial arrangement of pixels. As mentioned in [64], in the case of HFUS images, the values found in the individual channels of the images are important. Based on the previously segmented masks, the pixels belonging to each skin layer were extracted.

In order to determine the textural features of the dermatoscopic images, they were decomposed into red, green, blue, and hue, saturation, and value channels (see Figure A1 in Appendix B). 

For both modalities, the following features were calculated: histogram-, gray level co-occurrence matrix (GLCM) [65], local binary patterns (LBP) [27]—and coefficients of scattered wavelet transform-based features [28]. For HFUS images, echogenicity features [49] were obtained by calculating the ratio of low echogenic pixels (LEP), medium echogenic pixels (MEP), and high echogenic pixels (HEP) to all pixels within the mask [49,66]. Values calculated for the SLEB were referenced to the dermis layer echogenicity. 

***Features extracted from neural network*:** Various publicly available, annotated image datasets [53,67] enable model training for dermoscopic image classification. Among models described as sufficient for image classification of combined HAM1000035 and ISIC 2020 [68] databases, the EfficientNet [69] was chosen as the most accurate in AK detection. The model was trained on the combined dermoscopic image database [53,68] to classify the images as AK and no AK cases. It was then used for feature extraction from dermoscopic images being analyzed in our study. The features from the last fully connected layer were considered for further analysis. The model focus was evaluated using the GradCAM algorithm [70]. The model training parameters are listed in Table A3, Appendix B.

#### 2.2.2. Assessment of the Stage of AK 

Due to the presence of outliers in the data, a z-score normalization was chosen to ensure robustness to such values [71]. The features were then ranked using the minimum redundancy and maximum relevance algorithm (MRMR) [72] for dermatoscopy, ultrasonography, and both combined. The optimal cut-off point for the relevant features was searched iteratively with a simultaneous search for the optimal values of the control parameters for the selected classifier. 

Due to differences in group sizes, the SMOTE oversampling technique of synthetic data generation for minority classes was used [73] for each training set. The new dataset includes existing data and modified samples, reinforcing information already existing for minority groups but not creating new signals for these groups [73]. 

The last step of the analysis was to determine whether it was possible to distinguish between different stages of AK based on the obtained characteristics. For this, the most promising in preliminary experiments, the support vector machine (SVM) [74] algorithm, was selected and trained using leave-one-out validation (each patient in a separate fold). The utilized SVM parameters are given in Appendix B (Table A4). 

## 3. Results

### 3.1. Segmentation

The HFUS image segmentation results were evaluated using the Dice index [75], which for the entry echo layer was over 0.95 and for the SLEB layer about 0.91. The accurate segmentation results ensure that the automated segmentation step does not influence further analysis. The classification results obtained were the same as for manually segmented regions. The exemplary segmentation results are included in Figure 2. 

### 3.2. Classification 

To analyze the influence of different features and different considered modalities, the classification was carried out consecutively on handcrafted features from HFUS, handcrafted from dermatoscopy, combined handcrafted and extracted using the neural network from dermatoscopy, and combined features from HFUS and dermatoscopy. 

The classification accuracies and unweighted Cohen’s kappa [76] values obtained for different feature combinations are summarized in Table 1. The corresponding confusion matrices are shown in Figure 3. The obtained results are interpreted according to the following rules [76,77,78]: values below 0 indicate no and 0.01–0.20 none to slight, 0.21–0.40 fair, 0.41–0.60 moderate, 0.61–0.80 substantial, and 0.81–1.00 almost perfect agreement.

### 3.3. Statistical Analysis 

The features extracted from HFUS and dermoscopic images can be divided into two categories. The first group consists of features that can be medically interpreted. The second group includes those features whose association with structures on the images is seemingly impossible, yet they improve the final classification accuracy. 

Interpretable features, which take part in the final decision, were then additionally subjected to statistical analysis, which enabled the identification of characteristics that differentiated between groups. The Kruskal–Wallis test [79] (alpha 0.05) was used to determine whether significant differences exist between groups. The effect size was calculated using eta squared value [79], and differences between groups were determined using Dunn’s post hoc test [79] with Bonferroni correction.

Statistical analysis of ultrasound features shows significant differences in median values of epidermal characteristics related to thickness and structure. The reported effect size was small but not negligible. Post hoc analysis shows differences between groups 1 and 3. The parameters describing the thickness and those corresponding to the roughness of the entry echo layer had higher median values as the degree of AK increased. Examples of entry echo-related features are presented in Figure 4a,b. For the SLEB layer, significant differences between groups were noted for the MEP and pixel intensity distribution (Figure 4c,d). 

For the texture features determined for the SLEB layer, contrast allows differentiation between groups 1 and 2, and homogeneity shows differences between group 1 and the others (see example in Figure 4e,f). Texture analysis for the combined entry echo and SLEB layers reveals that for homogeneity and GLCM correlation, there are significant differences between group 1 and the others (Figure 4g,h). In the case of these features, a medium effect size was recorded.

## 4. Discussion

Several noninvasive techniques were proposed to facilitate diagnosis in the assessment of AK, including clinical presentation, dermatoscopy, HFUS, and reflectance confocal microscopy [80]. For the first time in this study, we used ML-based algorithms to determine the usefulness of HFUS and dermatoscopic features in AK staging. The obtained classification results indicate that the AI-based method can be helpful for AK assessment. The features obtained from the EfficientNet model trained to differentiate AK from other skin tumors strongly improve the AK stage assessment results. Interestingly, features that enable the distinction of AK from other lesions can be used to assess the degree of AK. 

The Cohen’s kappa values collected in Table 1 indicate slight agreement with the council of experts in the analysis of HFUS and dermatoscopic data described by handcrafted features. Using features extracted from the EfficientNet model increased the accuracy to over 81% and Cohen’s kappa value to 0.73, indicating substantial agreement. Combining the NN model and handcrafted HFUS features resulted in the second score (accuracy 79%, kappa 0.71). The accuracy was lower. However, the HFUS features were indicated as important in MRMR analysis, which combined dermatoscopic neural network- and HFUS handcrafted descriptors. Therefore, further analysis of HFUS images utilizing the NN model for feature extraction should be beneficial. 

The limited number of training examples included in this study made the use of deep convolutional models at each step of the analysis impossible. However, increased image data in future work would increase the obtained accuracy and enable replacing the SVM classifier with a deep neural network, too. To further prove the usability of HFUS in automated AK staging, more HFUS images are desirable. Further improvement can also be seen for lower class imbalance. Additional research, including various devices (HFUS scanners and dermatoscopes), will make the results independent of the measurement method. 

Despite the relatively low accuracy of the classifier trained using HFUS handcrafted features caused by a lack of features differentiating all the three AK stages, their statistical analysis shows important features that can be useful to distinguish separate pairs. 

Typically, AKs are graded clinically according to a classification introduced by Olsen et al. [81], based on an assessment of the thickness of the lesion and the presence of scales.

This is also concomitant with the image analysis findings in this study—the entropy within the pictures depended on the staging—and might be associated with the clinically visible and palpable roughness corresponding to the overproduction of keratin [82]. In the HFUS image of AK, the atypical keratinocytes are responsible for reducing skin echogenicity, similar to other skin tumors. In AKs, usually decreased echogenicity, sometimes forming linear SLEB, can be visualized. Due to the accumulation of the scales (keratin) on the AK surface, perpendicular to the entry echo, shadows may be observed, making depth assessment challenging. However, from a clinical point of view, it is important to differentiate AK from SCC, which may present in HFUS similarly to AK, although as a deeper low echogenic mass, suggesting invasion. It should be emphasized that SLEB may correspond to elastosis and concomitant inflammation, not only tumor formation, making interpreting the US image challenging. Firstly, SLEB was detected in photodamaged skin [51,83,84], where it corresponds to elastosis, accumulation of glycosaminoglycans and tissue water in the papillary dermis [51,84,85], and its thickness might correlate with the severity of photodamage [85]. Moreover, over the years, SLEB has also been described in inflammatory disorders such as psoriasis and atopic dermatitis [8,86,87] or skin lymphomas [87] as a helpful treatment follow-up parameter [88,89]. 

In our analysis, lesions clinically assessed as AK 1 revealed more contrast than AK 2 and AK 3 in HFUS, which may suggest additional features responsible for decreased echogenicity. In a Fernandez-Figueras et al. [90] study, most of the SCCs developed on AK background consisted of overlaying AK 1 in the biopsy. We hypothesize that the difference in contrast between the scans might correspond to the inflammation, which is usually invisible on dermatoscopy but might be seen in HFUS as a decreased echogenicity. On histology, the AK base commonly shows chronic inflammation ranging from scattered lymphoid cells to dense infiltration with lichenoid features [90,91]. Berhane [92] proposed a clinically identified inflammatory phase model, being a transitional phase to SCC, where a clinically recognizable inflammation was found prior to the invasion stage. Moreover, in some recent studies, the presence of elastosis in sun-damaged skin was considered a form of stromal atrophy and inflammation in field cancerization and, subsequently, assumed to precede the tumor formation [93].

Therefore, we suspect that HFUS may be a complementary method to dermatoscopy to assess the risk of progression. The limitation of this finding is that we did not evaluate the lesions histopatologically, as the level of histopathological dysplasia does not correlate with the clinical assessment, and we cannot be sure whether the decreased echogenicity corresponds to inflammation, tumor formation, or elastosis [94]. The aim of this study was a noninvasive evaluation of AK. If there are no doubts clinically and dermoscopically, the diagnosis of this entity does not require a biopsy. However, further studies on correlations between HFUS and histopathological examination are planned.

In our analysis, the entry echo in HFUS differed between AK stages and increased within the dermatoscopic and clinical grading. The entry echo corresponds to only dead epidermal layers in HFUS, not the whole epidermis. Nevertheless, as published by Heerfordt et al. [95], the thickness of the epidermis does not predict the epidermal dysplasia or p53 expression in the examined lesions; in fact, in another study by Heerfordt et al. [93], most of the pre-examined AKs with severe dysplasia presented a thin epidermis. 

We proposed combining a neural network model with a support vector machine for multimodal image analysis (dermatoscopy and HFUS). In our view, this combination may facilitate the staging of AKs and may be beneficial in clinical practice in predicting the risk of progression to malignancy. The value of HFUS relies on additional features that are not detected during dermatoscopic and clinical examination (i.e., the analysis of subclinical lesions) and, therefore, enables us to select patients who require a prompt therapeutic approach. We suspect that the use of AI in HFUS might also be applicable in field cancerization, which is very often present in patients with multiple AKs [96]. Since the entire skin is exposed to ultraviolet radiation, subclinical changes corresponding to the presence of elastosis and atypical cells are reported [97]. However, this would require further studies. 

## 5. Conclusions

In the era of the increasing role of AI in clinical practice, the application of its utility in HFUS is still evolving. Our study starts the pathway to exploring AI application in this noninvasive method, especially in dermato-oncology, in which HFUS has low specificity. We designed the ML and deep model-based image processing framework for multimodal data analysis and tested it on a cohort of 54 patients diagnosed with AK. Based on the obtained results, we concluded that the neural network models trained on public dermatoscopic image datasets can be adapted for AK assessment, and features that enable the distinction of AK from other lesions can be used to assess the degree of AK. Since the HFUS features were indicated as statically important in AK staging, their further combination with models utilizing HFUS images should improve the classification results. We believe that combining dermoscopy and HFUS altogether might be especially beneficial in complex cases with extensive sun damage or high-risk patients with a history of skin cancer. Moreover, from a future perspective, it may help determine the right spot for a diagnostic biopsy. Further studies on bigger groups are required to improve the algorithm, and a histopathological examination is needed to correlate the scans with the findings seen throughout the skin layers in AK and field cancerization. 

## Figures and Tables

**Figure 1 jcm-13-07499-f001:**
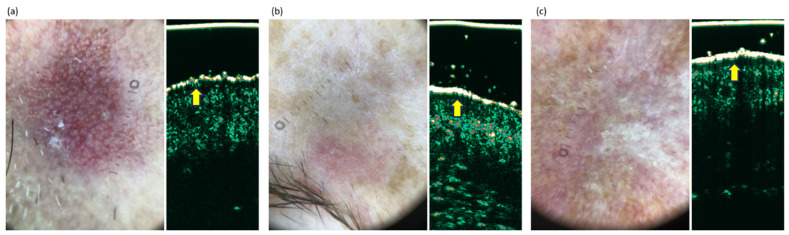
Exemplary image pairs (dermatoscopic image and high frequency ultrasound) recorded at the lesion site. The clinico-dermatoscopic classification of each AK was based on a three-point Zalaudek scale: (**a**) AK 1, (**b**) AK 1, (**c**) AK 2. The dermatoscopic images were acquired with DermLite DL5, 10× magnification coupled to a smartphone camera, and sized 3024 × 4032 pixels. The HFUS images of AKs were recorded with DermaScan C, a Cortex Technology device, linear 20 MHz probe, and sized 1024 × 224 pixels. Subepidermal low echogenic band (SLEB) seen beneath the entry echo in HFUS (indicated by the arrows).

**Figure 2 jcm-13-07499-f002:**
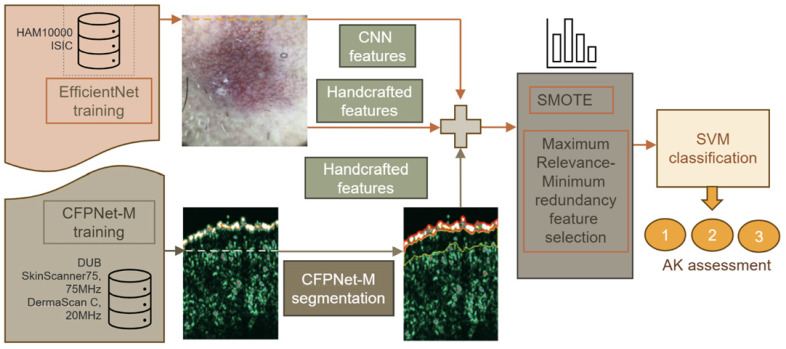
AI framework for multimodal image processing in AK assessment. Individual frames contain image modalities, types of extracted features, ML algorithms (MRMR and SVM for final classification and AK assessment 1–3), and deep neural network models (EfficientNet for dermatoscopic feature extraction—upper path, and CFPNet-M for HFUS image segmentation—lower path) applied at each analysis step. In places where due to insufficient training data it was impossible to use deep models, the handcrafted features are extracted.

**Figure 3 jcm-13-07499-f003:**
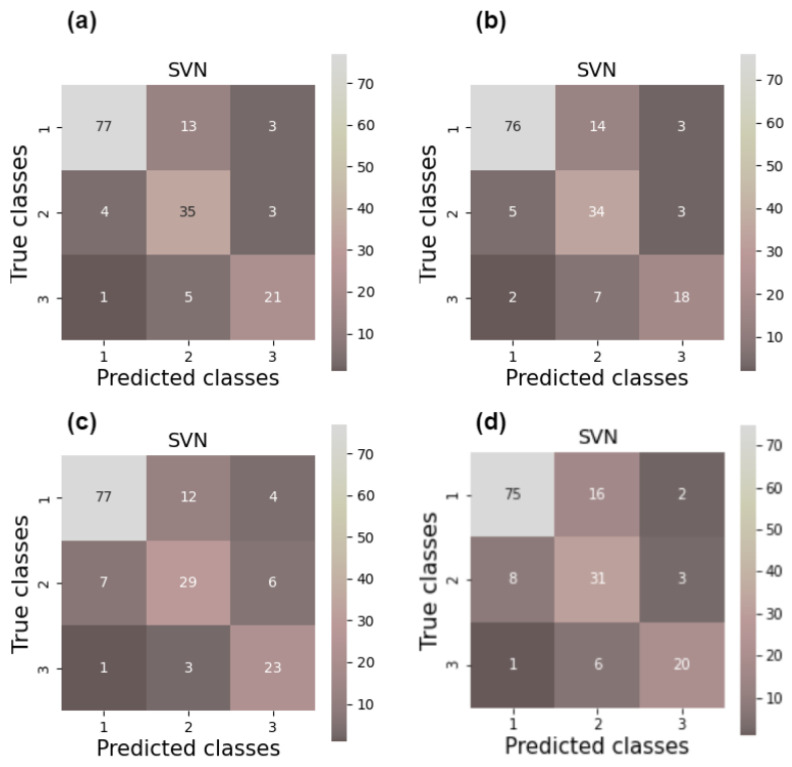
Confusion matrices for classification using different feature combinations: (**a**) dermatoscopy NN features, (**b**) dermatoscopy handcrafted features, and NN features, (**c**) HFUS handcrafted and dermatoscopy NN features, (**d**) HFUS handcrafted, dermatoscopy handcrafted, and NN features. Numbers refer to samples classified into each class.

**Figure 4 jcm-13-07499-f004:**
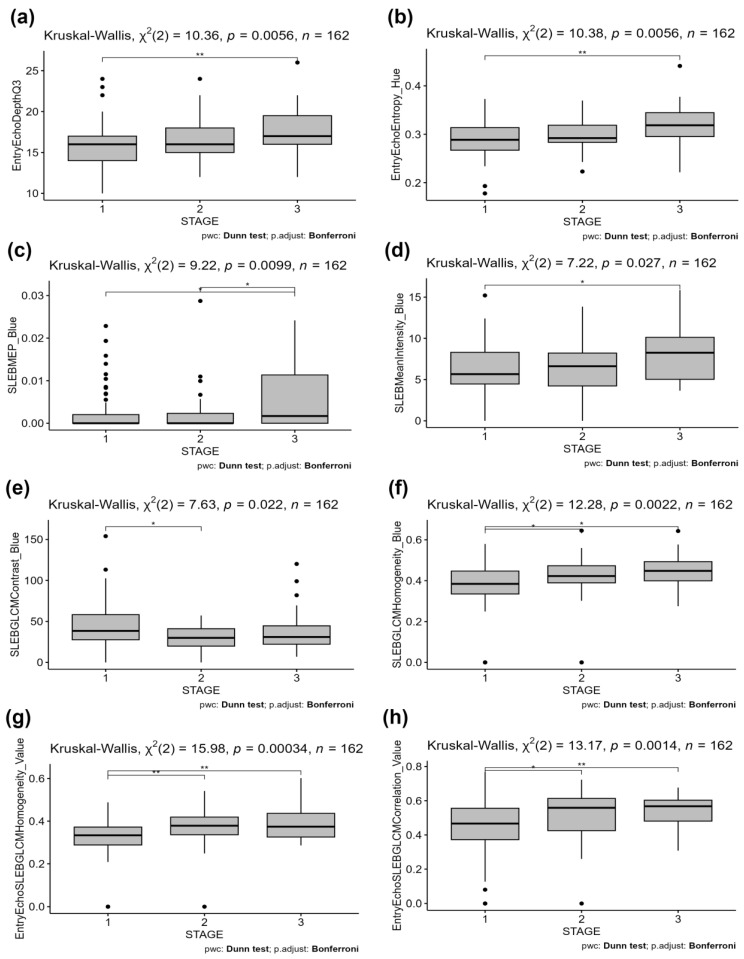
Statistically important features: (**a**) entry echo thickness 3rd quartile (effect size 0.05), (**b**) entropy of pixels in entry echo layer (effect size 0.05), (**c**) ratio of MEP in SLEB and dermis (effect size 0.05), (**d**) ratio of mean intensity in SLEB and dermis (effect size 0.03), (**e**) GLCM contrast in SLEB (effect size 0.04), (**f**) GLCM homogeneity in SLEB (effect size 0.06), (**g**) GLCM homogeneity for combined skin layers (effect size 0.09), (**h**) GLCM correlation for combined skin layers (effect size 0.07). Statistically significant differences between groups are marked with stars (*—*p* < 0.05, **—*p* < 0.01).

**Table 1 jcm-13-07499-t001:** Obtained classification accuracy, number of features used for final AK assessment and Cohen’s kappa value. The best obtained results were highlighted in color.

Features	Accuracy	Number of Features	Cohen’s Kappa
HFUS and dermatoscopy, handcrafted features	0.4833	11	0.1830
Dermatoscopy, NN ^1^ features	0.8130	42	0.7378
Dermatoscopy, handcrafted, and NN ^1^ features	0.7645	42	0.6847
HFUS, handcrafted, and dermatoscopy, NN ^1^ features	0.7901	45	0.7064
HFUS, dermatoscopy, handcrafted, and NN ^1^ features	0.7618	52	0.6860

^1^ NN—neural network.

## Data Availability

The data that support the findings of this study are available from the corresponding author upon reasonable request.

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
