# Peer review of "Automatic Assessment of AK Stage Based on Dermatoscopic and HFUS Imaging—A Preliminary Study"

_jcm, 2024, doi:10.3390/jcm13247499_

Round 1
Reviewer 1 Report
Comments and Suggestions for Authors
The algorithms you developed can facilitate accurate diagnosis, supporting clinicians in treatment planning. This study's future opportunities would be to approach a wider database to generalize the results and even apply them to other lesions.
Correlation with histopathological findings is important in every case; therefore, I would recommend a more ample discussion of the limitations. Can you also provide more information regarding the Fitzpatrick profile of the patients included?
Reviewer 2 Report
Comments and Suggestions for Authors
I reviewed the manuscript with the title “Automatic assessment of AK stage based on dermatoscopic and 2 HFUS imaging with the use of AI algorithms - a preliminary 3 study”
#Abstract of the study
Line 13- rewrite sentence “in current study..
Line 15- Methods- state the source/clinic/consortium of the patients
Line 25- with the use “of”…
Line 28, 29- hypothesise comes earlier. Please remove “we hypothesise”. May write further experiments are warranted with …..
#Introduction
Line 42- write full form of MMs and NMSCs
Line 52- SLEB
Please state clearly how the proposed experiments/work will help address the clinical need.
Also explain what is AI method. Is there an algorithm?
#Materials and methods:
Please provide details on how these patients were recruited under this study. Ethics in place? How the patients were identified to be recruited? What’s the patient recruitment criteria? What’s the patient selection criteria for this study? What is the control population if appropriate here? Where was this work carried out?
#Results:
Figure 1,2,3, 1B legends m=ned to be more descriptive with all the details.
Figures need to be embedded in the results section where appropriate.
#Discussion
Line 252- rephrase “slight agreement”
Line 315- re-phrase “we assume”
Line 318- please explain what is field cancerization?
The end of the discussion should be more on the conclusion of the study itself and how it can be improved, if needed, with more experiment or additional patient recruitment or images. Please also add how it will benefit the clinical set up? For example- rapid/accurate diagnosis, reduce burden on clinic, early detection etc.
The AI method needs to be discussed/explain better. Explain algorithm used if any, how it works, why is it better etc.
#Conclusion
Couldn’t find conclusion (unless I am missing any section of the manuscript)
#references
Need reviewing for reference style consistency
Overall although interesting the manuscript needs further improvement including English language accuracy and scientific writing.
Comments on the Quality of English Language
English language must be improved along with scientific writing necessary for research articles
Round 2
Reviewer 1 Report
Comments and Suggestions for Authors
Thank you for the changes provided. I believe the article can be accepted in its current form.
Author Response
Dear Reviewer,
Thank you for your positive review and appreciation of our revisions. We are delighted that you find the updated manuscript suitable for publication.
Regards,
The Authors
Reviewer 2 Report
Comments and Suggestions for Authors
Thank you for revising the manuscript.
Some comments:
#Abstract of the study
Line 10- Actinic keratosis (AK) - K capital
Line 11- fair skin type- what’s the Fritzpatrick scale, use scientific/medical terminology
Line 22- public datasets
Line 26- rephrase the sentence with correct font
Line 28- rephrase the sentence
#Introduction
Line 33- AK
Line 62- Machine learning (ML)- L capital
#Materials and methods:
Study design- in this prospective study cohort, were the patient consented for research use of the images captured and ethics approval etc. Can be stated in one sentence.
Fig 1 and 2: the legend should contain all the information about the patient, the grade of the lesion/patch, at what stage the images were captured, which technique was used, how many images were taken, are the images shown here representative of 3-4 images for example, what magnification was use (if appropriate) etc. What are green, red, yellow cells in the images, was any dye used?
This information is there in lines 120-126 for figure 1. I needs to be stated in legend as figure legends should be independent of the material/methods.
Same applies for figure 2 legend.
Also explain how AI was used in the methods- how it helped in the study, imaging, image analysis etc.
Is algorithm added in the manuscript? If not what are the restrictions? Can outline the algorithm for criteria used for imaging/processing etc.
There is only a line or two with the mention of AI in methods section.
#Discussion-
Line 396- combining a neural network model
#Conclusion- needs to be re-stated as conclusion rather than short description of the study.
English language – can be improved
Comments on the Quality of English Languagethe quality of English language can be improved.
